# The Different Clinicopathological Features of Remnant Gastric Cancer Depending on Initial Disease of Partial Gastrectomy

**DOI:** 10.3390/cancers12102847

**Published:** 2020-10-02

**Authors:** Won Ho Han, Bang Wool Eom, Hong Man Yoon, Young-Woo Kim, Myeong-Cherl Kook, Keun Won Ryu

**Affiliations:** 1Center for Gastric Cancer, National Cancer Center, Goyang 10322, Korea; 13408@ncc.re.kr (W.H.H.); kneeling79@ncc.re.kr (B.W.E.); red10000@ncc.re.kr (H.M.Y.); gskim@ncc.re.kr (Y.-W.K.); 2Department of Cancer Control and Population Health, National cancer Center Graduate School of Cancer Science and Policy & Center for Gastric Cancer, Goyang 10302, Korea

**Keywords:** remnant gastric cancer, partial gastrectomy, pyloric metaplasia, intestinal metaplasia, pathogenesis

## Abstract

**Simple Summary:**

Clinicopathological features and immunohistochemical subtype were evaluated in patients with remnant gastric cancer considering the initial cause of partial gastrectomy. In the malignant group, the proportion of male patients was substantially lower, with a shorter interval between initial partial gastrectomy and remnant gastric cancer. In the background mucosa around the carcinomas, incidence of high-grade pyloric metaplasia was significantly higher in the benign group (13/20, 65.0% vs. 10/28, 35.7%), while high-grade intestinal metaplasia was only observed in the malignant group (0/20, 0% vs. 7/28, 25.0%). The pathogenesis of remnant gastric cancer may differ depending on the initial disease. Benign disease was prevalently associated with pyloric metaplasia. Malignant disease was also accompanied by intestinal metaplasia.

**Abstract:**

*Background*: The incidence of gastric cancer increases in the remnant stomach after partial gastrectomy; however, its pathogenesis remains controversial. The clinicopathological features and immunohistochemical subtype were evaluated in patients with remnant gastric cancer considering the initial cause of partial gastrectomy. *Methods*: We categorized 59 cases of remnant gastric cancer who underwent curative surgery between 2001 and 2016 according to initial pathologies of benign (*n* = 24) or malignant (*n* = 35). Histological changes including pyloric metaplasia and intestinal metaplasia in the mucosa around the anastomosis site and the background mucosa of carcinomas were compared between the groups. *Results*: In the malignant group, the proportion of male patients was substantially lower, with a shorter interval. In background mucosa around the carcinomas, incidence of high-grade pyloric metaplasia was significantly higher in the benign group (13/20, 65.0% vs. 10/28, 35.7%), while high-grade intestinal metaplasia was only observed in the malignant group (0/20, 0% vs. 7/28, 25.0%). *Conclusions*: The cancers in the initial benign disease are mainly associated with pyloric metaplasia at the anastomosis site, reflecting reflux, but not with intestinal metaplasia. On the other hand, in the initial malignant disease group, intestinal metaplasia has an equally important role as reflux-associated pyloric metaplasia.

## 1. Introduction

Early diagnosis of gastric cancer according to national screening programs and advanced surgical treatment have led to an increase in the five-year survival rates in Korea and Japan [1,2,3]. Gastric resection with lymph node dissection is the only curative treatment for these patients [4]. Metachronous cancer in the remnant stomach after partial gastrectomy has gained great interest due to the increased application of organ function-preserving surgery [5].

The hypothesized risk factors for remnant gastric cancer include *Helicobacter pylori (H. pylori)* infection, changes in the gastric mucosa due to bile reflux, and the type of anastomosis methods [6,7,8]. However, most of these studies are retrospective and disease incidence and pathogenesis were not investigated. Several studies have reported that the changes in gastric mucosa caused by bile reflux after subtotal gastrectomy may increase the prevalence of remnant gastric cancer. This was in contrast to the results of other studies reporting that bile reflux may decrease *H. pylori* infection and generated a controversy about the relevant risk factors [9,10]. In order to understand the risk factors and features of remnant gastric cancer, combined pathological analysis of the anastomosis site and background mucosa is required. However, to date, very few studies have addressed this issue. 

Recent studies have suggested new molecular classifications of gastric cancer, aimed at improving prognosis and treatment [11,12]. Presumably, a better understanding of the changes occurring in the anastomosis mucosa, as well as molecular subtype classification based on immunohistochemistry, will shed light on the pathogenesis of remnant gastric cancer.

In this study, patients diagnosed with remnant gastric cancer who had undergone surgery were classified based on the initial disease after partial gastrectomy. The analysis of clinicopathological features and mucosal changes, as well as the immunohistochemical evaluation of molecular subtypes, were carried out to investigate the pathogenesis of remnant gastric cancer. 

## 2. Materials and Methods

### 2.1. Inclusion Criteria and Clinical Characteristics

Patients who underwent surgery for remnant gastric cancer at the National Cancer Center, Korea, from 2001 to 2016 were divided into a benign and a malignant group based on the reason for partial gastrectomy. In the case of remnant gastric cancer from residual disease of initial surgery, the recurrence within 5 years of surgery was excluded. In order to compare histologically in the same manner as the malignancy group, cases with anastomosis at an antrum on previous surgeries were excluded. Remnant gastric cancer was diagnosed by endoscopy and computed tomography (CT) prior to surgery. Ethical approval for the research protocol was provided by the institutional review board of the National Cancer Center (No. NCC2018-0207). The need for and patients’ informed consent was waived with approval of Institutional Review Board given the retrospective nature of the study.

The following clinical factors were evaluated: sex, age, *H. pylori* infection status, interval between initial partial gastrectomy and remnant gastric cancer surgery, initial surgical method, stage of malignancy at initial partial gastrectomy, pathological stage after surgery, histology, and Lauren classification. 

### 2.2. Pathological Analysis

To evaluate the cumulative effects of the reflux, foveolar hyperplasia (FH) [9] and pyloric metaplasia (PM) [13] were examined in the anastomosis area. In order to selectively focus on PM caused by reflux, excluding the effect of antralization [14], only the part without intestinal metaplasia (IM) was considered as PM. Based on PM distribution, the PM grade in the anastomosis area was determined for the zone within 3 cm from the anastomosis junction (see the Results section). 

To identify preneoplastic changes possibly related to carcinogenesis, the pathological features of the background mucosa surrounding the carcinomas were analyzed. The background mucosa was classified as normal-type when both PM and intestinal metaplasia were grade 0 or 1, as PM-type when PM was grade 2 or 3, and as intestinal metaplasia-type when intestinal metaplasia was grade 2 or 3.

The grade of metaplasia in the anastomosis area or the background mucosa was assigned using criteria of the Updated Sydney system. Briefly, grading is performed using the four-tiered grading system (0, 1, 2, 3) and the cut-off points of grade 1(≤30%), grade 2 (31–60%), and grade 3 (≥61%) [9,15].

### 2.3. Immunohistochemistry Analysis

The Epstein-Barr virus status, microsatellite instability, mucin phenotype, as well as the expression of E-cadherin, aberrant P53, aberrant HER-2, aberrant MET, and CDX2, were compared between the benign and the malignant group (Figure 1). Immunohistochemistry results were segregated based on the molecular The Cancer Genome Atlas (TCGA) classification [12,16].

### 2.4. Statistical Analysis

Continuous variables were presented as mean ± standard error and categorical variables were expressed as proportions. Categorical variables were compared using the Pearson χ^2^ test, while continuous variables were compared using the t-test or Mann–Whitney U test, as appropriate. The threshold for statistical significance was set at *p* < 0.05. All statistical analyses were performed using SAS version 9.1.3 for Windows (SAS Institute, Cary, NC, USA).

## 3. Results

### 3.1. Patient Demographics

In total, 59 patients were included in this study. Of these, 24 (most of whom had undergone ulcer surgery) were assigned to the benign group, and 35 were included in the malignant group, mainly after surgery for gastric adenocarcinoma. The malignant group included a lower number of male patients and was characterized by a shorter time interval between initial partial gastrectomy and remnant gastric cancer diagnosis. The cases of *H. pylori* infection were more abundant in the benign group, but the difference was not statistically significant (62.5% vs. 31.4%, *p* = 0.17) (Table 1).

### 3.2. Distribution of Reflux-Induced Mucosal Changes in the Anastomosis Area

For 41 cases, sections of non-neoplastic mucosa in the anastomosis area were available for analysis. FH and PM, which are hallmarks of reflux gastritis, were observed in most cases (37/41 and 38/41, respectively). Two changes occurred simultaneously within one gland, with FH on the upper layer and PM on the lower layer (Figure 1). Pyloric-type glands frequently descended to the submucosal layer and formed gastritis cystica profunda. Infiltration of inflammatory cells was very low. FH was distributed up to 5 cm from the anastomosis junction; in 1/3 of the cases it was distributed within 1.29 cm, whereas in 2/3 of the cases it was distributed up to 2.2 cm from the anastomosis junction. PM was distributed up to 5 cm from the anastomosis junction; in 1/3 of the cases it was distributed within 0.8 cm, whereas in 2/3 of the cases it was distributed within 1.9 cm from the anastomosis junction (Table 2). Based on these distribution patterns, we defined the severity within the area from the anastomosis junction to 3 cm as the grade of PM in the anastomosis area. 

### 3.3. Differences in the Mucosa of the Anastomosis Area between Benign and Malignant Cases

Tumor classification based on disease severity at the time of initial partial gastrectomy revealed a substantial difference in the anastomosis area. In the benign group, the tumors were located within the distribution distance of PM, whereas in the malignant group, the tumors were located beyond the PM range. While most patients in the benign group exhibited high-grade PM (18/19, 94.7%), those in the malignant group prevalently displayed low-grade PM (14/22, 63.6%). Most cases in both groups presented no or mild-grade intestinal metaplasia in the anastomosis area and moderate or higher grades were rarely observed (2/41) (Table 3). 

### 3.4. Differences in the Background Mucosa around the Carcinoma between Benign and Malignant Cases 

For 48 cases, sections of the background mucosa were available for analysis. A significant difference was noted between the groups (*p* = 0.028). In the benign group, most cases exhibited PM (13/20, 65.0%), while intestinal metaplasia (IM) was absent (0/20). The remaining cases exhibited normal mucosa (7/20, 35%). The malignant group exhibited all three mucosal types at a similar frequency (PM: 10/28, 35.7%; intestinal metaplasia: 7/28, 25.0%; normal mucosa: 11/28, 39.3%) (Table 4).

### 3.5. Results of Immunohistochemistry and Molecular Classification

In total, 52 cases were analyzed by immunohistochemistry. Molecular classification according to immunohistochemistry confirmed a significant prevalence of the Epstein-Barr virus (EBV) type in the benign group (7/22, 31.8% vs. 1/33, 3.0%). Moreover, a relatively high frequency of aberrant P53 (6/22, 27.3% vs. 10/33, 30.3%), as well as another type (9/22, 42.9% vs. 19/33, 57.6%) was observed in the malignant group, albeit the latter differences were not statistically significant. CDX2 expression was predominant in the malignant group (11/22, 50.0% vs. 26/33, 78.8%; *p* = 0.014) (Table 5).

## 4. Discussion

Numerous studies have described the clinical features of remnant gastric cancer. However, the pathogenesis and cause of the disease remains controversial. Moreover, little is known regarding the histological features of the anastomosis site and background mucosa in remnant gastric cancer. The present study investigated the clinical features of remnant gastric cancer based on initial disease of partial gastrectomy. In addition, histological analysis and molecular classification based on immunohistochemistry were conducted to investigate the pathogenesis of remnant gastric cancer. Remnant gastric cancers that occurred following benign disease were strongly associated with PM of the anastomosis site, whereas malignant remnant gastric cancers were characterized by both PM and intestinal metaplasia.

In accordance with previous studies, a significant difference was observed between the benign and the malignant group in the duration of the interval between the initial partial gastrectomy and the occurrence of remnant gastric cancer. Numerous studies have hypothesized that the shorter interval in the malignant group is related to the presence of a precancerous lesion, such as atrophic gastritis and intestinal metaplasia [17,18]. 

Reflux gastritis from bile may occur in the remnant stomach and its major histological feature is known as FH without inflammation [9]. In the present study, we analyzed the distribution and severity of reflux gastritis in detail. PM, in addition to FH, was observed around the anastomosis site. These characteristics of PM were frequently observed in the remnant stomach. On the other hand, the literature focusing on PM in the remnant stomach is extremely poor [13]. PM is a phenomenon by which fundic-type glands transform into mucin-secreting pyloric-type glands, and has been reported to occur in two forms. One is “antralization”, in which PM occurs in the fundic glands of the angle of stomach and gradually progresses toward the proximal part. As a result, the border of the fundic and pyloric mucosa (fundic-pyloric border) also moves toward the proximal part [19]. This phenomenon is associated with *H. pylori* gastritis and is usually accompanied by IM [14]. The other mechanism is “spasmolytic polypeptide-expressing metaplasia (SPEM),” which occurs separately as a spot within the fundic mucosa region, arises from chief cell transdifferentiation, and further progresses into IM [20]. Whether these two conditions are related remains unclear. PM observed in the present study exhibited distinct characteristics from the two aforementioned mechanisms. First, it started from the anastomosis site of the remnant stomach after partial gastrectomy. Second, it was accompanied by FH, which is a characteristic of reflux gastritis. Third, it was not accompanied by IM. 

We demonstrated that PM was the main preneoplastic change in the benign group. A previous study also reported PM in the remnant stomach. However, the relationship between PM and the initial disease status was not clarified, nor was the time interval from initial surgery evaluated [13]. In the latter study, only the presence or absence of PM was recorded, but its grade was not taken into consideration. In the present study, we demonstrated that although PM was present in almost all examined cases, its degree of severity was much higher in the benign group. High-grade PM was prevalent in the mucosa around the carcinomas in the benign group, while its incidence was much lower in the malignant group. This suggested that reflux-induced changes had a stronger impact on the carcinogenic background in the benign group than in the malignant group. 

Notably, high-grade IM was only observed in the mucosa of the malignant cases. High-grade IM is the result of mucosal changes due to prolonged *H. pylori* gastritis. Therefore, *H. pylori* gastritis and reflux-induced gastritis could have a similar influence on carcinoma development in the malignant group. It can be assumed that partial gastrectomy in the benign group was performed before *H. pylori* gastritis could progress into the body and, therefore, background mucosa had no chance to develop.

Significant differences in EBV and CDX2 expression were found between the two groups, as determined by immunohistochemistry. EBV-type gastric cancer was highly represented in the benign group (31.8%), while it was only present in only 3% of the malignant cases, which is lower than the average incidence among total gastric cancers (5–10%) [21,22]. EBV-type gastric cancer exhibits unique molecular alterations [16]. However, the mechanism of its development is not well-established. EBV-type gastric cancer is highly prevalent in the cardia and body among remnant gastric cancers, but it has no relationship with *H. pylori* infection [23,24]. In a recent study, the EBV infection rate was higher in patients with remnant gastric cancer compared with those with conventional gastric cancer. It has been reported that Billroth II anastomosis, carcinoma at the anastomosis site, diffuse type, and EBV genome polymorphisms are related to the EBV-associated remnant gastric cancer [25]. In the present study, EBV-type gastric cancer was higher in the benign group compared to the malignant group, and this may be related to the Billroth II anastomosis in all benign groups. CDX2 was poorly expressed in the benign group, and this could be related to the low grade of IM in the background mucosa of benign cases [26]. CDX2 is a homeobox gene expressed in the intestinal mucosa and has a role in the differentiation of intestinal epithelium [27]. CDX2 is expressed in all gastric carcinomas that develop an intestinal phenotype, but it is also present in a proportion of cases with gastric phenotype [28]. The clinical relevance of the different molecular subtypes of gastric carcinoma is not clear at present, but carcinogenic factors might differ depending on the initial disease.

In the malignant group, remnant gastric cancer occurs after 5 years or more, and at the time of diagnosis, about 50% is stage II or higher (Appendix A). Annual endoscopic surveillance after gastrectomy in the malignant group should be recommended in at least 10 years, rather than the 5- year follow-up. Meanwhile high-grade pyloric metaplasia was mainly distributed within 3 cm in the benign group (Appendix A). Considering the data of the present study, it could be suggested to carefully observe the anastomosis area (within 3 cm) in the endoscopic surveillance 10 years from the partial gastrectomy performed for the benign cause. (Figure 2)

The main limitations of this study are its retrospective design and the relatively small number of cases; the examined sections of some cases were unavailable for the histological analysis of the anastomosis site. In addition, the pathogenic role of background mucosa would be more accurately analyzed if only small-sized cancers were included for analysis. These issues need to be addressed in future studies. 

## 5. Conclusions

In conclusion, remnant gastric cancers following benign disease were prevalently associated with PM of the anastomosis site, whereas those developing after malignant disease were also accompanied by IM. The pathogenesis of remnant gastric cancer may differ depending on the initial disease for partial gastrectomy.

## Figures and Tables

**Figure 1 cancers-12-02847-f001:**
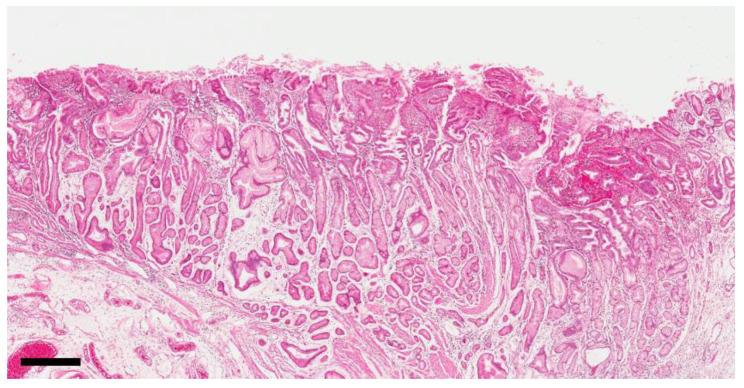
Reflux-induced changes in the anastomosis area. The upper epithelial layer showed foveolar hyperplasia and the lower glandular layer showed pyloric metaplasia in the absence of intestinal metaplasia. (magnification 40×). Scale bar represents 500 µm.

**Figure 2 cancers-12-02847-f002:**
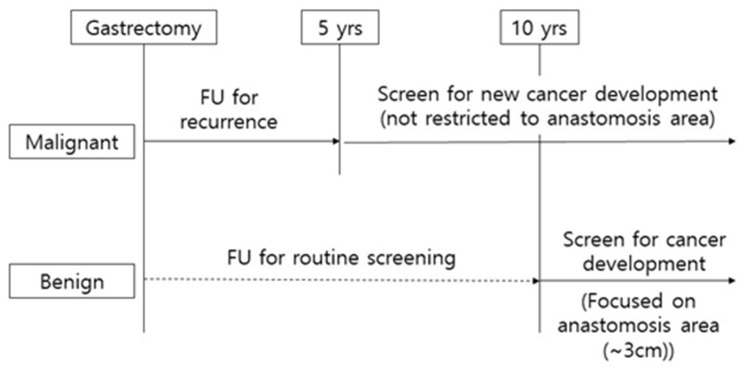
Scheme of clinical surveillance for remnant gastric cancer.

**Table 1 cancers-12-02847-t001:** Patient demographic.

Category	Variables	Benign (*n* = 24)	Malignancy (*n* = 35)	*p* Value
Age (years)		63.9 ± 8.7	65.1 ± 11.5	0.66
Sex	Male	24 (100%)	27 (77.1%)	0.02
	Female	0 (0%)	8 (22.9%)	
Interval (years)		30.1 ± 8.0	13.4 ± 6.8	<0.001
Initial operation	Billroth I	0 (0%)	7 (20%)	0.06
	Billroth II	24 (100%)	28 (80%)	
Pathologic stage of initial adenocarcinoma	I	-	24 (68.6%)	
	II		3 (8.6.%)	
	III		1 (2.9%)	
	IV		0 (0%)	
	unknown		7 (20.0%)	
Pathologic Stage of remnant cancer	I	13 (54.2%)	19 (54.3%)	0.71
	II	6 (25.0%)	10 (28.6%)	
	III	4 (16.7%)	6 (17.1%)	
	IV	1 (4.2%)	0 (0%)	
LN metastasis	Negative	15 (62.5%)	25 (71.4%)	0.47
	Positive	9 (37.5%)	10 (28.6%)	
Tumor size		4.6 ± 2.1	4.4 ± 2.4	0.52
Histology	Differentiated	15 (62.5%)	16 (45.7%)	0.2
	Undifferentiated	9 (37.5%)	19 (54.3%)	
Lauren classification	Intestinal	14 (58.3%)	12 (34.3%)	0.07
	Diffuse	8 (33.3%)	17 (48.6%)	
	Mixed	2 (8.3%)	6 (17.1%)	
*H. pylori* infection	Negative	8 (33.3%)	21 (60.0%)	0.17
	Positive	15 (62.5%)	11 (31.4%)	
	Unknown	1 (4.2%)	3 (8.6%)	

**Table 2 cancers-12-02847-t002:** Distribution range of reflux-induced mucosal changes from the anastomosis junction.

Variables	Median	Maximum	33 Percentile	67 Percentile
Foveolar hyperplasia Pyloric metaplasia	1.3 cm 1.5 cm	5 cm 5 cm	1.29 cm 0.8 cm	2.2 cm 1.9 cm

**Table 3 cancers-12-02847-t003:** Tumor location and the mucosal characters of anastomosis area.

Variables	Grade	Benign (*n* = 19)	Malignancy (*n* = 22)	*p*-Value
Distance from tumor center to anastomosis (cm)		1.80 ± 0.57	2.98 ± 0.53	0.14
Length of PM *		2.3 ± 0.27	0.85 ± 0.15	<0.001
PM grade	Low (grade 0,1)	1 (5.3%)	14 (63.6%)	<0.001
	High (grade 2,3)	18 (94.7%)	8 (36.4%)	
IM * grade	Low (grade 0,1)	19 (100%)	20 (91%)	0.927
	High (grade 2,3)	0	2 (9%)	

* PM: pyloric metaplasia, IM: intestinal metaplasia.

**Table 4 cancers-12-02847-t004:** Differences between the types of background mucosa surrounding carcinomas.

Mucosal Type	Benign (*n* = 20)	Malignancy (*n* = 28)	*p*-Value
Normal mucosa Pyloric metaplasia Intestinal metaplasia	7 (35.0%) 13 (65.0%) 0 (0%)	11 (39.3%) 10 (35.7%) 7 (25.0%)	0.028

**Table 5 cancers-12-02847-t005:** Classification based on immunohistochemistry and in situ hybridization.

Variables	Classification	Benign (*n* = 22)	Malignancy (*n* = 33)	*p* Value
Molecular type	EBV type	7 (31.8%)	1 (3.0%)	0.068
	MSI type	0	3 (9.1%)	
	E-cadherin aberrant type	1 (4.5%)	1 (3.0%)	
	P 53 aberrant type	6 (27.3%)	10 (30.3%)	
	Other type *	9 (42.9%)	19 (57.6%)	
HER-2 aberrant	Negative	22 (100.0%)	29 (87.9%)	0.106
	Positive	0 (0.0%)	4 (12.1%)	
MET aberrant	Normal	12 (54.5%)	23 (66.7%)	0.404
	Abnormal	10 (45.5%)	10 (33.3%)	
Mucin phenotype	Gastric type	7 (31.8%)	10 (30.3%)	0.827
	Gastric-intestinal type	5 (22.7%)	10 (30.3%)	
	Intestinal type	4 (18.2%)	6 (18.2%)	
	Null type	6 (27.2%)	7 (21.3%)	
CDX2 expression	Negative	11 (50.0%)	7 (21.2%)	0.014
	Positive	11 (50.0%)	26 (78.8%)	

* Other type: Normal P53 expression with lack of Epstein-Barr virus (EBV), microsatellite instability (MSI) deficiency, and E-cadherin aberrant.

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
