# Peer review of "The Different Clinicopathological Features of Remnant Gastric Cancer Depending on Initial Disease of Partial Gastrectomy"

_cancers, 2020, doi:10.3390/cancers12102847_

Round 1

Reviewer 1 Report

The paper reports a clinical relevant problem. The series is interesting and should be complemented with other series from other centres. Taking the data presented the authors should propose a strategy to decrease the number of tumour recurrence. In fact, the paper can be much more appealing if authors propose a pipeline of patient surveillance.  Therefore, I suggest that authors propose a clinical surveillance pipeline in the discussion session that other clinicians can follow or refute.

The paper is mainly descriptive and the authors using the data obtained need to include indications how these type of patients should be monitor after gastrectomy. In fact, the big drawback of the paper is the lack of a scheme or pipeline of clinical surveillance of the different groups: benign and malignant. That is my major concern. It is not a matter of detail is a matter of clinical implications of the data.
Cancers is a highly renowned journal that should require that the papers should have clinical value is this type of research.

Author Response

September 25, 2020

Manuscript ID: cancers-943220

Title: The different clinicopathological features of remnant gastric cancer depending on initial disease of partial gastrectomy

We thank Professor Ms. Avery Liu and the reviewers of Cancers for a positive response on our manuscript. We would like to express our sincere gratitude to the reviewers and editor for their time and effort in reviewing our manuscript. Each comment has helped improve our manuscript in various respects. Please find herewith our responses to the reviewers’ comments.

Reply to Reviewer.

The paper reports a clinical relevant problem. The series is interesting and should be complemented with other series from other centres. Taking the data presented the authors should propose a strategy to decrease the number of tumour recurrence. In fact, the paper can be much more appealing if authors propose a pipeline of patient surveillance. Therefore, I suggest that authors propose a clinical surveillance pipeline in the discussion session that other clinicians can follow or refute.

The paper is mainly descriptive and the authors using the data obtained need to include indications how these type of patients should be monitor after gastrectomy. In fact, the big drawback of the paper is the lack of a scheme or pipeline of clinical surveillance of the different groups: benign and malignant. That is my major concern. It is not a matter of detail is a matter of clinical implications of the data.

Cancers is a highly renowned journal that should require that the papers should have clinical value is this type of research.

  • High grade pyloric metaplasia was mainly distributed within 2 cm in the benign group. Considering the data of the present study, it could be suggested to carefully observe the anastomosis area (within 2 cm) in the endoscopic surveillance 10 years from the partial gastrectomy performed for the benign cause.

Thank you for your consideration. We look forward to hearing from you.

Sincerely,

Keun Won Ryu, MD, PhD

Center for Gastric Cancer, Research Institute & Hospital, National Cancer Center

323 Ilsan-ro, Ilsandong-gu, Goyang-si 410-769, Republic of Korea.

Myeong-cherl Kook, MD, PhD

Center for Gastric Cancer, Research Institute & Hospital, National Cancer Center

323 Ilsan-ro, Ilsandong-gu, Goyang-si 410-769, Republic of Korea

 Fax; 82-31-920-0069

Reviewer 2 Report

Interesting and well written study on different clinicopathological features of remnant gastric cancer in a specific population of patients undergone to gastric surgery. The cohort analysed is quite small, but the method of study precise and well described are the results in relation to the classification of different immunohistochemical and pathological/molecular elements. Moreover, authors underlined that the study considered a small number of cases, but this could be a prerequisite for future larger studies.

In the discussion the controversial role that Epstein-Barr virus (EBV) may play as a causal role in the pathogenesis of gastric remnant carcinoma might be deepened, in relation to recent literature.

My comment is in relation to a deeper mention in the Discussion on the issue of the correlation between EBV infection and remnant stomach cancer, referring to the literature.

Author Response

September 25, 2020

Manuscript ID: cancers-943220

Title: The different clinicopathological features of remnant gastric cancer depending on initial disease of partial gastrectomy

We thank Professor Ms. Avery Liu and the reviewers of Cancers for a positive response on our manuscript. We would like to express our sincere gratitude to the reviewers and editor for their time and effort in reviewing our manuscript. Each comment has helped improve our manuscript in various respects. Please find herewith our responses to the reviewers’ comments.

Reply to Reviewer.

Interesting and well written study on different clinicopathological features of remnant gastric cancer in a specific population of patients undergone to gastric surgery. The cohort analysed is quite small, but the method of study precise and well described are the results in relation to the classification of different immunohistochemical and pathological/molecular elements. Moreover, authors underlined that the study considered a small number of cases, but this could be a prerequisite for future larger studies.

In the discussion the controversial role that Epstein-Barr virus (EBV) may play as a causal role in the pathogenesis of gastric remnant carcinoma might be deepened, in relation to recent literature.

My comment is in relation to a deeper mention in the Discussion on the issue of the correlation between EBV infection and remnant stomach cancer, referring to the literature.

  • In a recent study, the EBV infection rate was higher in patients with remnant gastric cancer compared with those with conventional gastric cancer. It has been reported that Billroth II anastomosis, carcinoma at the anastomosis site, diffuse type, and EBV genome polymorphisms are related to the EBV-associated remnant gastric cancer. In the present study, EBV-type gastric cancer was higher in the benign group compared with the malignant group, and this may be related to the Billroth II anastomosis in all benign groups.

Thank you for your consideration. We look forward to hearing from you.

Sincerely,

Keun Won Ryu, MD, PhD

Center for Gastric Cancer, Research Institute & Hospital, National Cancer Center

323 Ilsan-ro, Ilsandong-gu, Goyang-si 410-769, Republic of Korea.

Myeong-cherl Kook, MD, PhD

Center for Gastric Cancer, Research Institute & Hospital, National Cancer Center

323 Ilsan-ro, Ilsandong-gu, Goyang-si 410-769, Republic of Korea

 Fax; 82-31-920-0069

Round 2

Reviewer 1 Report

The authors were asked to implement in the paper a pipeline for surveillance of the patients to increase the interest of the journal readers. In fact, this was the single major comment related to the paper.  The authors have introduced some extra information along the text but did not made the effort to design a comprehensive scheme for both situations of disease monitoring and treatment, in benign and malignant situations.  This request is mandatory for acceptance to make the information clinically relevant.

Author Response

September 28, 2020

Manuscript ID: cancers-943220

Title: The different clinicopathological features of remnant gastric cancer depending on initial disease of partial gastrectomy

We thank Professor Ms. Avery Liu and the reviewers of Cancers for a positive response on our manuscript. We would like to express our sincere gratitude to the reviewers and editor for their time and effort in reviewing our manuscript. Each comment has helped improve our manuscript in various respects. Please find herewith our responses to the reviewers’ comments.

Reply to Reviewer.

Reviewer 1

The authors were asked to implement in the paper a pipeline for surveillance of the patients to increase the interest of the journal readers. In fact, this was the single major comment related to the paper.  The authors have introduced some extra information along the text but did not made the effort to design a comprehensive scheme for both situations of disease monitoring and treatment, in benign and malignant situations.  This request is mandatory for acceptance to make the information clinically relevant.

  • We added the Scheme of clinical surveillance for remnant gastric cancer (Figure 2). we also added supplement figures about cumulative incidence of remnant gastric cancer and scatter plot according to initial disease.
  • We described it in Discussion : In the malignant group, remnant gastric cancer occurs after 10 years or more, and at the time of diagnosis, about 50% is stage II or higher. Annual endoscopic surveillance after gastrectomy in malignant group should be recommended at least 10 years, rather than the 5 years follow-up. Meanwhile high grade pyloric metaplasia was mainly distributed within 3 cm in the benign group. Considering the data of the present study, it could be suggested to carefully observe the anastomosis area (within 3 cm) in the endoscopic surveillance 10 years from the partial gastrectomy performed for the benign cause.”

Thank you for your consideration. We look forward to hearing from you.

Sincerely,

Keun Won Ryu, MD, PhD

Center for Gastric Cancer, Research Institute & Hospital, National Cancer Center

323 Ilsan-ro, Ilsandong-gu, Goyang-si 410-769, Republic of Korea.

Myeong-cherl Kook, MD, PhD

Center for Gastric Cancer, Research Institute & Hospital, National Cancer Center

323 Ilsan-ro, Ilsandong-gu, Goyang-si 410-769, Republic of Korea

 Fax; 82-31-920-0069